# Synthetic Peptides Containing Three Neutralizing Epitopes of Genotype 4 Swine Hepatitis E Virus ORF2 induced Protection against Swine HEV Infection in Rabbit

**DOI:** 10.3390/vaccines8020178

**Published:** 2020-04-13

**Authors:** Yiyang Chen, Tianxiang Chen, Yuhang Luo, Jie Fan, Meimei Zhang, Qin Zhao, Yuchen Nan, Baoyuan Liu, En-Min Zhou

**Affiliations:** 1Department of Preventive Veterinary Medicine, College of Veterinary Medicine, Northwest A&F University, Yangling 712100, China; chenyiyang@nwsuaf.edu.cn (Y.C.); ctx1995@nwafu.edu.cn (T.C.); 2018050532@nwsuaf.edu.cn (Y.L.); fj199597@nwsuaf.edu.cn (J.F.); ZMM0320@nwafu.edu.cn (M.Z.); qinzhao_2004@nwsuaf.edu.cn (Q.Z.); nanyuchen2015@nwsuaf.edu.cn (Y.N.); 2Scientific Observing and Experimental Station of Veterinary Pharmacology and Diagnostic Technology, Ministry of Agriculture, Yangling 712100, China

**Keywords:** hepatitis E virus, swine HEV, neutralizing epitopes, synthetic peptides

## Abstract

Genotype 4 hepatitis E virus (HEV) is a zoonotic pathogen transmitted to humans through food and water. Previously, three genotype 4 swine HEV ORF2 peptides (^407^EPTV^410^, ^410^VKLYTS^415^, and ^458^PSRPF^462^) were identified as epitopes of virus-neutralizing monoclonal antibodies that partially blocked rabbit infection with swine HEV. Here, individual and tandem fused peptides were synthesized, conjugated to keyhole limpet hemocyanin (KLH), then evaluated for immunoprotection of rabbits against swine HEV infection. Forty New Zealand White rabbits were randomly assigned to eight groups; groups 1 thru 5 received three immunizations with EPTV-KLH, VKLYTS-KLH, PSRPF-KLH, EPTVKLYTS-KLH, or EPTVKLYTSPSRPF-KLH, respectively; group 6 received truncated swine HEV ORF2 protein (sp239), and group 7 received phosphate-buffered saline. After an intravenous swine HEV challenge, all group 7 rabbits exhibited viremia and fecal virus shedding by 2–4 weeks post challenge (wpc), seroconversion by 4–9 wpc, elevated alanine aminotransferase (ALT) at 2 wpc, and severe liver lymphocytic venous periphlebitis. Only 1–2 rabbits/group in groups 1–4 exhibited delayed viremia, fecal shedding, seroconversion, increased ALT levels, and slight liver lymphocytic venous periphlebitis; groups 5–6 showed no pathogenic effects. Collectively, these results demonstrate that immunization with a polypeptide containing three genotype 4 HEV ORF2 neutralizing epitopes completely protected rabbits against swine HEV infection.

## 1. Introduction

Hepatitis E virus (HEV) infects approximately 20 million people worldwide each year, causing 3.3 million cases of acute illness and 30,000–40,000 deaths annually [1,2]. Generally, HEV is self-limiting, but mortality rates in pregnant women can approach 25% in HEV-endemic areas [3,4], while chronic HEV infection adversely impacts immunosuppressed individuals [5,6]. In addition to detrimental HEV effects on human health and survival, HEV outbreaks also negatively impact societal and economic health [7]. Consequently, public health agencies should focus resources on controlling HEV infection through the implementation of effective public sanitation measures worldwide.

HEV belongs to the family *Hepeviridae*, which includes two genera, *Orthohepevirus* and *Piscihepevirus* [8]. *Orthohepevirus* includes four HEV species (A to D), of which *Orthohepevirus A, C*, and *D* species mainly infect mammalian hosts, while the *Orthohepevirus B* species mainly infects chickens [8]. Within the *Orthohepevirus A* species, HEV genotypes 1 and 2 are restricted to humans [9], while genotypes 3 and 4 are zoonotic, with main reservoirs of deer, pigs, and wild boars. The latter two hosts have prompted the use of the designation “swine” HEV for genotypes 3 and 4 [10]. Indeed, in China, genotype 4 is the main HEV genotype found in pig herds [11,12]. As the first identified animal HEV, swine HEV shares 80% to 90% nucleotide sequence identity with human HEV isolates from the same geographic regions [13]. Recent work conducted in China suggests that genotype 4 HEV infectious virus can potentially enter the human food chain through meat [12]; therefore, blocking the spread of HEV within pig herds is particularly important.

HEV is a quasi-enveloped, positive-sense, single-stranded RNA virus [8,14] that contains three open reading frames (ORFs), ORF1, ORF2, and ORF3 [15]. ORF2 encodes the viral capsid protein, which is approximately 660 amino acids (aa) in length. The capsid protein contains immunodominant and neutralizing epitopes of virus particles and is also the target of protective humoral immune responses [16,17]. To date, two conformational and four linear neutralizing epitopes have been identified within the ORF2 protein of genotype 1 HEV [18]. Most of these epitopes (with one exception) are located mainly within the capsid protein E2 domain, spanning aa 459 to 602 [19]. Interestingly, although all four major HEV genotypes share a single serotype, a monoclonal antibody (MAb) recognizing a conformational neutralizing epitope was only able to neutralize genotype 1 HEV [20,21]. These results suggest that slight structural diversity exists among ORF2 proteins of the four major HEV genotypes. Notably, few published reports have described antibody-neutralizing epitopes residing within genotype 4 swine HEV ORF2 protein [22].

Previous reports had demonstrated rabbits to be susceptible to experimental infection by genotype 4 human and pig HEV isolates [23,24,25], prompting the use of rabbits as an animal model for studying HEV pathogenicity. Importantly, a rabbit model has many advantages [23,26], including lower cost and easier manipulation as compared with large animal models. Therefore, rabbits were used here for in vivo MAb neutralization assays.

Meanwhile, other reports have demonstrated that a bacterially expressed peptide fragment (designated p239) spanning the region of genotype 1 HEV capsid protein between aa 368 and aa 606 could self-assemble into virus-like particles (VLPs). This self-assembling peptide was subsequently further developed into the first vaccine against genotype 1 HEV [27,28]. Here we strived to create a vaccine against genotype 4 swine HEV based on our previous detection of three neutralizing epitopes recognized by three novel MAbs (1B5, 2C7, and 2G9) that bound to capsid protein of genotype 4 swine HEV [22]. Of these MAbs, 2C7 and 2G9 could partially inhibit genotype 4 swine HEV infection of HepG2 cells in vitro and of rabbits in vivo. In addition, MAb 1B5, which recognizes an epitope common to HEV isolates that infect various host species (including birds), also showed partial neutralizing ability both in vitro and in vivo [16,22]. These results infer that peptides recognized by these MAbs may themselves generate naturalizing anti-HEV antibodies in vivo after immunization. Therefore, here three peptides of genotype 4 swine HEV capsid protein neutralizing epitopes (EPTV, VKLYTS, and PSRPF) and tandemly fused peptides (EPTVKLYTS and EPTVKLYTSPSRPF) were synthesized and tested for their ability to prevent genotype 4 swine HEV infection in vivo. These results provide a foundation for the future development of HEV vaccines and antiviral drugs.

## 2. Materials and Methods

### 2.1. Synthesis Peptides, sp239, and ORF3 Protein and Virus

Three neutralizing epitopes, ^407^EPTV^410^, ^410^VKLYTS^415^, and ^458^PSRPF^462^, were characterized (Figure 1), and corresponding peptides were synthesized then analyzed for protective capacity, as were tandem epitopes of EPTVKLYTS and EPTVKLYTSPSRPF. Each peptide was conjugated either to keyhole limpet hemocyanin (KLH) for immunization or to bovine serum albumin (BSA) for ELISA detection. All peptides used in this study were synthesized by GenScript (Nanjing, China) and were diluted according to the manufacturer’s recommendations.

Swine HEV truncated ORF2 protein sp239 (aa 368-606) and ORF3 protein were expressed separately in *Escherichia coli* BL21 (DE3) host cells and purified as described previously [22].

Swine HEV (strain CHN-SD-sHEV, genotype 4, GenBank accession no. KF176351) was isolated from a bile sample of a 32-week-old pig at a slaughterhouse in China. Viral stock containing 10^4^ genomic equivalents (GE)/mL was prepared from suspensions of fecal and bile samples collected from specific-pathogen-free (SPF) pigs infected with CHD-SD-sHEV, as previously described [16].

### 2.2. Immunization of Rabbits with Synthetic Peptides of Three Neutralizing Epitopes of Genotype 4 Swine HEV ORF2

A total of 40 8-weeks-old SPF New Zealand White rabbits (Chengdu Dossy Laboratory Animal Technology Co., Ltd., Chengdu, China) with initial body weights averaging 1.5 kg (SD, 1.74 kg) were assigned to eight groups (5 rabbits per group). Rabbits were housed separately in 40 cages. Before immunization, each rabbit was screened to confirm serum negativity for anti-HEV antibodies and fecal negativity for HEV RNA using indirect ELISA and nested RT-PCR, respectively.

After one week of acclimation, 9-week-old rabbits of groups 1–6 were immunized subcutaneously with Pep VKLYTS, Pep PSRPF, Pep EPTV, Pep EPTVKLYTS, Pep EPTVKLYTSPSRPF, or sp239, respectively. Rabbits in group 7 were immunized as above, but with PBS only (negative control). Nonimmunized and unchallenged rabbits in group 8 served as normal controls. The antigen (400 μg) for each immunization was mixed with Imject^®^ Alum (Thermo Scientific, USA, 77161), and two additional injections were at 2-week intervals.

### 2.3. Virus Challenge and Sample Collection

Rabbits in groups 1–7 were each challenged by injection via ear vein of 1 mL of swine HEV stock (10^4^ GE/mL) prepared from a 10% fecal stock. Rabbit serum and fecal samples were collected before inoculation and weekly thereafter for 12 weeks. Viremia and fecal virus shedding were determined for each inoculated rabbit using a sensitive nested RT-PCR assay, while serum HEV RNA levels were concurrently tested using real-time RT-PCR. In addition, serum samples were tested for anti-swine HEV antibodies by indirect ELISA using swine HEV ORF3 (sHEV-ORF3) as coating antigen, and serum levels of alanine aminotransferase (ALT) were also measured. During necropsy, liver tissues were collected for routine histological examination.

### 2.4. Indirect ELISA

Before and after immunization and challenge, indirect ELISAs were used to measure levels of antibodies specific for peptides, sp239, or sHEV-ORF3 proteins in serum samples collected from rabbits previously immunized with BSA-conjugated antigens (peptides, sp239 protein, or sHEV ORF3 protein), as previously described by Chen et al. [22]. Briefly, ELISA plates were coated with 200 ng/well of various peptides, sp239 protein, or ORF3. After blocking and wash steps, serum samples (100 μL/well) were added into each well and incubated for 1 h at 25 °C. After three washes, horseradish peroxidase (HRP)-conjugated goat anti-rabbit IgG (Jackson ImmunoResearch, West Grove, PA, USA) diluted 1:4000 (100 μL/well) was added into wells then plates were incubated for 1 h at 25 °C. After three washes, 3,3′,5,5′-tetramethylbenzidine (TMB) was added to wells, and plates were incubated in the dark for 15 min at 25 °C. The colorimetric reaction was stopped by adding 3 M H_2_SO_4_ (50 μL/well), and optical density (OD) values were read at 450 nm using an automated microplate reader (Bio-Rad, USA). Each serum sample was tested in duplicate wells. The OD values of the pre-inoculation serum samples were used to determine cutoff values.

### 2.5. Qualitative and Quantitative Detection of Swine HEV RNA Using RT-PCR

Swine HEV RNA was detected in fecal and serum samples using RT-nPCR according to the method described by Huang et al. [29]. In addition, swine HEV RNA in serum samples was quantified using qRT-PCR methods, as previously described [30].

### 2.6. Determination of Serum Alanine Aminotransferase (ALT) Levels

ALT concentrations of serum samples collected weekly were measured (Hitachi 912; Roche, Indianapolis, IN, USA) following the manufacturer’s instructions. Rabbits were considered positive for hepatitis if post-challenge ALT levels exceeded pre-challenge ALT levels by greater than two-fold [31].

### 2.7. Gross and Microscopic Hepatic Lesions

During necropsies, gross pathological lesions in the liver of each rabbit were evaluated. Liver tissues were also fixed in 10% neutral buffered formalin and processed for routine histological examination.

### 2.8. Statistical Analysis

*t*-tests (Microsoft Office Excel 2016) were performed to compare differences in ELISA absorbance values reflecting levels of antibodies (specific for epitopes within peptides and sp239 antigens) in pre-vaccinated versus post-vaccinated rabbit sera; antibody levels are predictive of immunoprotection conferred by immunization with peptides or sp239 protein versus that of the PBS-immunized group. ALT values among groups were also analyzed using the same method [32]. *p* < 0.05 was considered significant.

### 2.9. Ethics Statement

All experiments in this study were designed based on the principles expressed in the Guidance for Experimental Animal Welfare and Ethical Treatment by the Ministry of Science and Technology of China. Experimental procedures and animal use and care protocols were carried out in accordance with the guidelines of the Northwest A&F University Institutional Committee for the Care and Use of Laboratory Animals and were approved by the Committee on Ethical Use of Animals of Northwest A&F University (AE124319).

## 3. Results

### 3.1. Immune Responses in Rabbits Immunized with Synthetic Peptides of Three Neutralizing Epitopes of Genotype 4 Swine HEV ORF2

Serum samples of each rabbit before and at various weeks post-immunization (wpi) were collected to measure levels of antibodies specific for peptides or sp239 protein using indirect ELISA. After the second immunization (4 wpi), the OD_450mm_ values of immunized rabbit serum samples (groups 1–6) increased to 3.0 and remained stable after the third immunization (6 wpi, Figure 2). Meanwhile, the antiserum collected from rabbits immunized with peptide EPTVKLYTS and peptide EPTVKLYTSPSRPF were also detected using the single peptide as coating antigen, the results showed there was strong antigen–antibody interaction, and no significant difference among each single peptide (*p* > 0.05, data not shown).

### 3.2. Measurements of Serum Antibody and HEV RNA Levels in Serum and Feces after Challenge

Before the challenge, serum and fecal samples from all rabbits were negative for HEV RNA (Figure 3). Conversely, five rabbits inoculated with the virus only (group 7) developed viremia and fecal shedding at 2 weeks post challenge (wpc), with seroconversion (antibody responses against ORF3 protein) observed at 4 wpc and lasting to 9 wpc (Table 1, Figure 3G7). No fecal virus shedding, viremia, or seroconversion were observed in rabbits immunized with peptide EPTVKLYTSPSRPF (group 5) or sp239 protein (group 6) throughout the experiment (Table 1, Figure 3G5,G6); these results infer that peptide EPTVKLYTSPSRPF and sp239 protein protected rabbits from swine HEV infection. However, in group 1, Rab1 (No. 1 rabbit) exhibited viremia and fecal virus shedding between 4 and 6 wpc with seroconversion observed from 6 to 9 wpc. Meanwhile, Rab4 showed transient viremia and fecal virus shedding at 5 wpc, with seroconversion observed at 9 wpc (Figure 3G1). In group 2, Rab8 and Rab10 developed transient fecal virus shedding at 5 and 6 wpc, respectively, with seroconversion lasting from 7 to 11 wpc or 5 to 7 wpc, respectively (Figure 3G2). In group 3, Rab14 showed fecal shedding at 8 wpc, with seroconversion lasting from 9 to 11 wpc (Figure 3G3). In group 4, Rab16 showed transient viremia and fecal virus shedding at 7 wpc, with seroconversion lasting from 8–11 wpc (Figure 3G4).

### 3.3. Quantitation of HEV RNA in Serum Samples

Besides the detection of HEV RNA in serum samples by RT-nPCR, viral copy numbers were quantitatively evaluated by RT-qPCR [30]. In group 7, the serum viral copy number peaked at 2–3 wpc (10^3.2^–10^4.2^ copies/mL) in all rabbits (Figure 4G7). Meanwhile, in groups 1 and 4, the highest viral copy numbers observed in serum samples of Rab1, Rab4, and Rab16 ranged between 10^3.1^ to 10^3.6^ copies/mL (Figure 4G1,G4), while peak numbers of viral copies were below 10^2^ copies/mL in all remaining rabbits of groups 1–6 (Figure 4). The protocol of the QuantTect probe RT-PCR Kit noted that 10^2^ copies/mL as a starting point (the fluorescence background values influenced the threshold).

### 3.4. ALT evaluation in Serum Samples

Throughout the study, ALT levels in serum samples of all rabbits were monitored weekly. In group 7, a peak ALT level of 99-113 U/L was observed for all rabbits at 2 wpc (Table 1, Figure 5G7). Conversely, no major ALT level changes were observed in groups 5 and 6 or in normal group 8 (Table 1, Figure 5G5, G6, and G8). However, in group 1, Rab1 and Rab4 showed increased ALT levels (102 and 96 U/L, respectively) in serum samples at 3 wpc (Figure 5G1); in group 2, increased ALT levels were observed for Rab8 and Rab10 (92 and 96 U/L, respectively) at 5 and 4 wpc, respectively (Figure 5G2); meanwhile, Rab14 (group 3) and Rab16 (group 4) developed increased respective ALT levels (96 and 98 U/L) at 4 wpc (Figure 5G3,G4, respectively); increased levels in these rabbits were delayed by 1–3 weeks as compared with group 7 rabbits (Table 1, Figure 5G1–G4). In addition, increases in serum ALT levels in group 7 rabbits were significantly greater than increases in group 8 sera at 2 wpc, but not at other wpc times (*p* < 0.05).

### 3.5. Gross and Microscopic Lesions

Throughout the study, no gross hepatic lesions were observed in livers of any necropsied rabbits in any group. However, slight to severe lymphocytic venous periphlebitis microscopic lesions were observed in liver tissues of some rabbits, including all group 7 rabbit livers (Table 1, Figure 6G7). By contrast, no microscopic liver lesions were observed in the livers of rabbits of groups 5, 6, and 8 (Table 1, Figure 6G5, G6, and G8, respectively). In addition, slight lymphocytic venous periphlebitis was observed in liver sections of 1 or 2 rabbits (Rab1, 4, 8, 10, 14, and 16) in other groups (Table 1, Figure 6G1b, G2b, G3b, and G4b), while no apparent microscopic liver lesions were observed in the remaining rabbits of groups 1–4 (Table 1, Figure 6G1a, G2a, G3a, and G4a).

## 4. Discussion

Genotype 3 and 4 HEVs are regarded as potential zoonotic viruses, with pigs serving as a main natural reservoir for virus propagation [33]. With the development of improved sanitation in China, the source of HEV for human outbreaks has moved from contaminated water to food sources [11]. Thus, it is important that human public health agencies work to block the spread of swine HEV in pig herds. Due to the lack of a highly efficient in vitro cell culture system, which has hampered traditional HEV vaccine development [34], genetic engineering of a subunit vaccine is currently being pursued. Our previous study identified three linear neutralizing B-cell epitopes in swine HEV ORF2 protein [22]. Therefore, in this study, synthetic peptides based on three neutralizing epitopes were used to immunize rabbits, followed by an assessment of immunoprotective effects against swine HEV infection. The results showed that peptides EPTV, VKLYTS, PSRPF, and tandem fused peptide EPTVKLYTS could partially protect rabbits against swine HEV infection, while immunization with tandem fused peptide EPTVKLYTSPSRPF or sp239 protein conferred complete protection.

Previous studies have pinpointed major immunodominant epitopes on the ORF2 protein that are candidate antigens for HEV vaccine development [35]. Based on analysis of the crystal structure of the ORF2 protein, there are three major capsid protein domains: shell (S) (aa 129 to 319), middle (M) (aa 320 to 455), and protruding (P) (aa 456 to 604) domains [19]. The P domain, commonly referred to as the E2s domain, forms a spike that acts as the predominant antigenic domain that contains all reported genotype 1 HEV neutralizing epitopes [18,36]. In this study, the peptide PSRPF within the P domain could protect rabbits from swine HEV infection, as did peptides EPTV and VKLYTS of the M domain. However, a previous study had reported that an epitope located within aa 403 to 417 of the M domain lacked neutralizing ability [37]. These results suggest that other neutralizing epitopes exist within the HEV ORF2 capsid protein besides the P domain, warranting additional research to more fully reveal the biological functions of various epitopes within the ORF2 protein.

It has been reported that immunization of SPF chickens with peptide Pep1B5-1, which includes the epitope (VKLYTS) recognized by MAb 1B5, was not protective against avian HEV challenge [16]. In the present study, peptide VKLYTS could partially inhibit swine HEV infection in rabbits, indicating that this peptide is a neutralizing epitope of swine HEV, as consistent with our previous results showing that MAb 1B5 could partially block swine HEV infection of rabbits. These results also confirm our previous speculation that the epitope in avian HEV may not be exposed on the virus surface, although further investigations are warranted to understand the nature of this epitope more fully.

Generally, slight to severe microscopic lesions in liver tissues were observed that demonstrated hepatitis was a complication of HEV infection, as indicated by lymphocytic venous periphlebitis, portal fibrosis, swollen hepatocytes, balloon-like lesions, and hepatocellular necrosis [23,25,38]. In our previous study, lymphocytic venous periphlebitis, balloon-like lesions, and hepatocellular necrosis were all observed in liver sections of CHN-SD-sHEV group rabbits [25]. However, here only severe lymphocytic venous periphlebitis was observed in CHN-SD-sHEV-inoculated group rabbit livers (Figure 6). The discrepancy in results may be due to different doses of virus or different time points between studies. In addition, immunoperoxidase staining of liver sections was performed, and results showed no antigens were detected (data not shown), which suggested that the infected animals may completely recover from the experimental infection, and the antigen is no longer detectable in liver at the endpoint of experiment. Therefore, in future studies, differences in microscopic liver lesion and immunoperoxidase staining results should be analyzed while also accounting for differences in dose and tissue sampling time points.

In general, 1–2 booster immunization were performed after the first immunization in the animal experiment. In this study, rabbits were immunized three times in total, and the antibody levels peaked at 4 wpi (after 2nd immune) and remained stable at 6 wpi (after 3rd immune) (Figure 2), which suggested that two times immunization can produce sufficient antibodies. If pigs were used as an animal model, two times immunization or one-time high dose immunization may provide adequate protection for pigs from HEV infection.

Importantly, one or two rabbits in groups immunized with peptides EPTV, VKLYTS, or PSRPF showed active infection but exhibited delayed fecal virus shedding and viremia as compared with the virus-only group (Figure 3). These results agree with those of our previous study that demonstrated that a single MAb protected rabbits from swine HEV infection [22]. The underlying reason for rare breakthrough infections may be due to inefficient neutralization achieved with a single epitope. Immunization with peptide EPTVKLYTSPSRPF, which comprises tandem epitopes recognized by MAbs 2G9, 1B5, and 2C7, provides the same protection against swine HEV in rabbits as achieved via immunization with sp239 protein, inferring that antibodies against these epitopes exhibited good neutralizing activities. Nevertheless, compared with peptides harboring single or double epitopes, the three tandem epitopes in peptide EPTVKLYTSPSRPF conferred the best immunoprotection, reflecting synergistic antiviral effects among three epitopes. Thus, further research is needed to develop this tandem peptide into an effective HEV vaccine against an increased virus dose that is protective for a prolonged period. Meanwhile, these epitopes are highly conserved in human, swine, and rabbit HEVs, bolstering their promise as HEV vaccine candidates with neutralizing activities against HEVs that infect different species.

## 5. Conclusions

In summary, rabbits immunized with peptide EPTVKLYTSPSRPF or sp239 protein showed no seroconversion, fecal virus shedding, viremia, ALT level increases, or liver lesions. This study revealed that peptides synthesized by tandem neutralizing epitopes effectively protected rabbits from swine HEV infection, while also providing a foundation for future vaccine and antiviral drug development.

## Figures and Tables

**Figure 1 vaccines-08-00178-f001:**
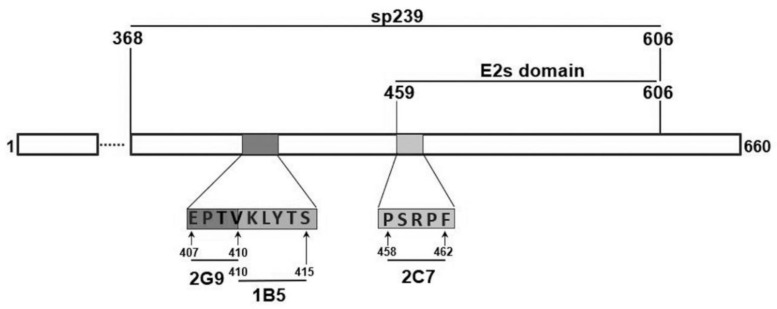
Schematic diagram showing linear positions of sp239, E2s domain, and genotype 4 swine hepatitis E virus (HEV) capsid protein-neutralizing epitopes recognized by monoclonal antibody (Mab) 2G9, 1B5, and 2C7.

**Figure 2 vaccines-08-00178-f002:**
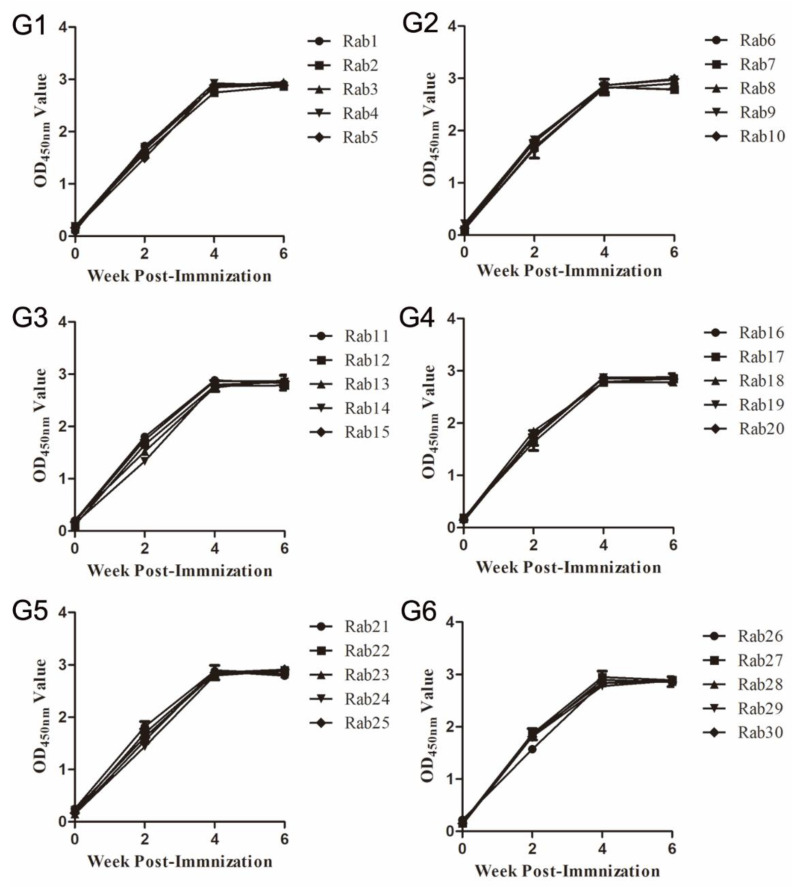
Seroconversion in rabbits immunized with Pep VKLYTS (**G1**), Pep PSRPF (**G2**), Pep EPTV (**G3**), Pep EPTVKLYTS (**G4**), Pep EPTVKLYTSPSRPF (**G5**), or sp239 (**G6**). Serum samples were collected from rabbits before immunization and 2 weeks after each immunization (1st, 2nd, 3rd, and 4th). ELISA plates were coated with 200 ng/well of various peptides or sp239 protein, and each point represents the OD_450nm_ value for each rabbit serum sample. There were statistical differences for the mean levels of antibodies in post-vaccinated versus pre-vaccinated rabbit sera in each group (n = 5, *p* < 0.05). The error bar indicates the confidence interval of reduplicates of three test wells.

**Figure 3 vaccines-08-00178-f003:**
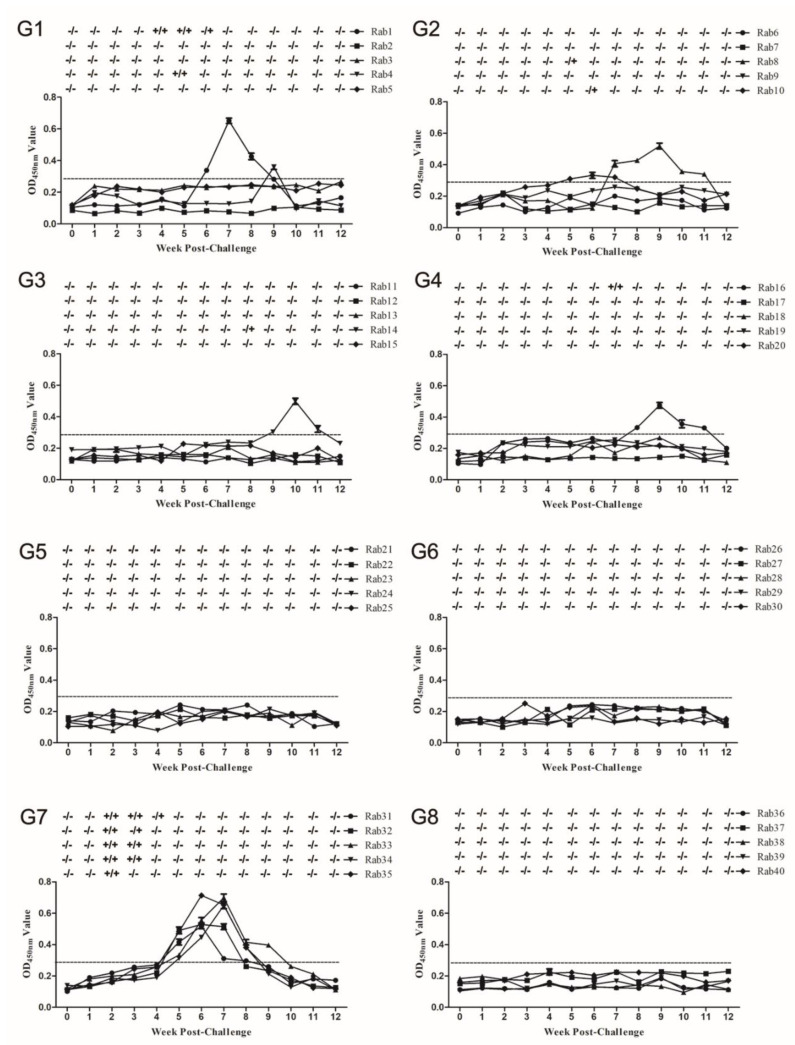
Time course of seroconversion, viremia, and fecal virus shedding in specific-pathogen-free (SPF) rabbits after challenge with CHN-SD-sHEV. SPF rabbits immunized with Pep VKLYTS (**G1**), Pep PSRPF (**G2**), Pep EPTV (**G3**), Pep EPTVKLYTS (**G4**), Pep EPTVKLYTSPSRPF (**G5**), sp239 protein (**G6**), or PBS ((**G7**), negative control); Rabbits challenged with PBS (**G8**) as normal control. HEV RNA was detected by RT-nPCR. Fecal or serum HEV RNA presence or absence are indicated by “+” and “−”, respectively. Serum and fecal samples were collected before virus inoculation and various weeks post challenge (wpc). The error bar indicates confidence interval of reduplicates of three test wells.

**Figure 4 vaccines-08-00178-f004:**
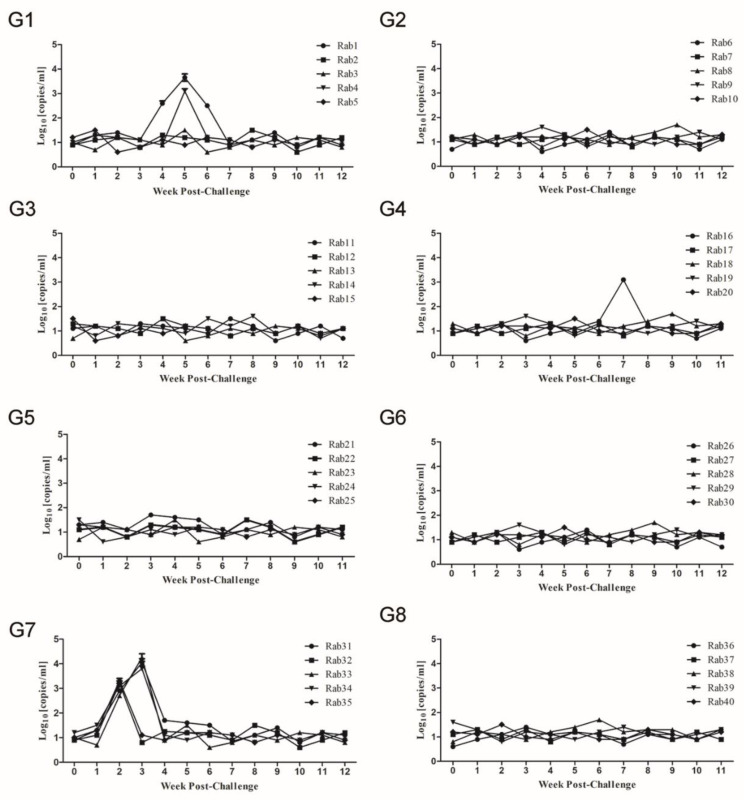
Dynamic changes in viral loads of serum samples from inoculated rabbits after challenge with CHN-SD-sHEV. SPF rabbits immunized with Pep VKLYTS (**G1**), Pep PSRPF (**G2**), Pep EPTV (**G3**), Pep EPTVKLYTS (**G4**), Pep EPTVKLYTSPSRPF (**G5**), sp239 protein (**G6**), or PBS ((**G7**), negative control); Rabbits challenged with PBS (**G8**) as a normal control. Serum samples were collected before virus inoculation and various weeks post challenge wpc. Y-axis represents log10 [copies/mL] of swine HEV RNAs in serum. The error bar indicates the confidence interval of reduplicates of three test wells.

**Figure 5 vaccines-08-00178-f005:**
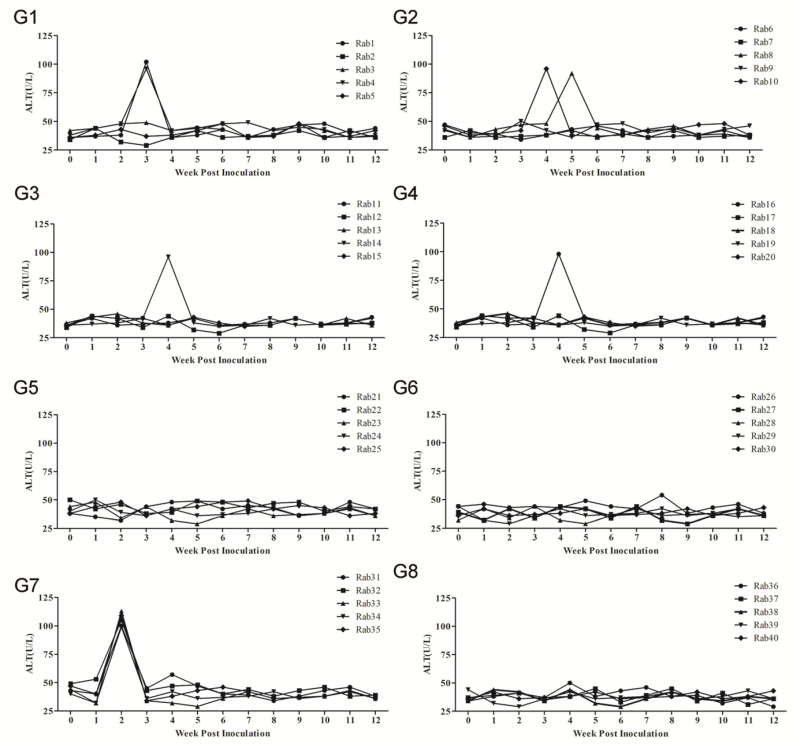
Levels of ALT liver enzyme in sera of inoculated rabbits in different groups. SPF rabbits immunized with Pep VKLYTS (**G1**), Pep PSRPF (**G2**), Pep EPTV (**G3**), Pep EPTVKLYTS (**G4**), Pep EPTVKLYTSPSRPF (**G5**), sp239 protein (**G6**), or PBS ((**G7**), negative control); Rabbits challenged with PBS (**G8**) as a normal control. Compared with the normal group 8, the mean aminotransferase (ALT) values from the 5 rabbits in group 7 were significantly higher at 2 wpi (n = 5, *p* < 0.05). *p*-values were calculated using Student’s *t*-test.

**Figure 6 vaccines-08-00178-f006:**
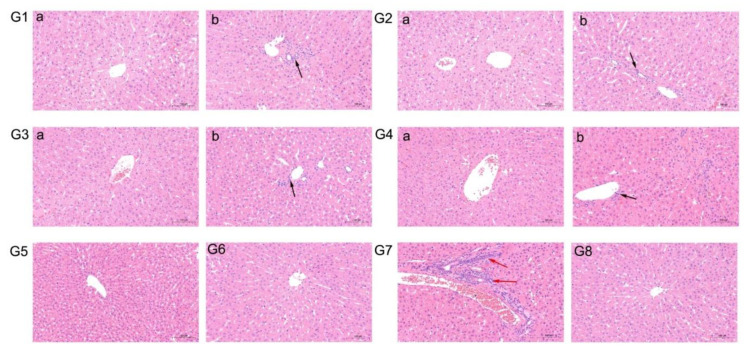
Various characteristics of microscopic lesions in livers from rabbits after challenged with CHN-SD-sHEV. (**G1a**–**G4a**, **G5**, **G6**, **G8**) No visible pathological signs of HEV infection in liver sections (Panels (**G1a**–**G4a**, **G5**, **G6**, **G8**)from Rab2, Rab6, Rab11, Rab17, Rab21, Rab26, and Rab36, respectively); (**G1b**)–(**G4b**) Slight lymphocytic venous periphlebitis observed in liver sections (Panels (**G1b**)–(**G4b**) from Rab1, Rab8, Rab14, and Rab16, respectively; black arrows); (**G7**) Severe lymphocytic venous periphlebitis in liver sections (Panel (**G7**) from Rab31; red arrow). Tissues were stained with hematoxylin and eosin.

**Table 1 vaccines-08-00178-t001:** Number of rabbits infected by CHN-SD-sHEV in different groups.

Group	Immunization	Seroconversion to Rabbit HEV	Fecal Virus Shedding	Viremia	ALT Increased	Microscopic Lesions
1	VKLYTS	4/5	2/5	2/5	2/5	2/5
2	PSRPF	3/5	3/5	0/5	2/5	2/5
3	EPTV	3/5	1/5	0/5	1/5	1/5
4	EPTVKLYTS	1/5	1/5	1/5	1/5	1/5
5	EPTVKLYTSPSRPF	0/5	0/5	0/5	0/5	0/5
6	Sp239	0/5	0/5	0/5	0/5	0/5
7	PBS (NC)	5/5	5/5	5/5	5/5	5/5
8	PBS (Normal)	0/5	0/5	0/5	0/5	0/5

The rabbits were infected as evidenced by seroconversion, fecal virus shedding, viremia, aminotransferase (ALT) increase, and microscopic lesions of livers. The number was shown as positive number/total number.

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
