# Peer review of "Synthetic Peptides Containing Three Neutralizing Epitopes of Genotype 4 Swine Hepatitis E Virus ORF2 induced Protection against Swine HEV Infection in Rabbit"

_vaccines, 2020, doi:10.3390/vaccines8020178_

Round 1

Reviewer 1 Report

Hepatitis E virus infects humans and other animals such as pigs. Swine HEV (sHEV) displays high genomic similarities to human HEV (hHEV) and therefore it represents a valuable animal model for studying human HEV infection. Also, HEV infection is zoonotic it is important prevent the infection in animals which are consumed by humans. The authors have recently characterized three neutralizing epitopes within the HEV capsid protein. Here, the authors have used the sHEW animal model to evaluate the above three neutralizing epitopes for their potency to elicit an anti-HEV immune response. The study demonstrates that a synthetic short peptide encompassing the three neutralizing epitopes is as effective as the truncated HEV ORF2 protein in eliciting an immune response to HEV. The manuscript flows well and would be of interest to a wide range of research groups including virologists and immunologists. Please see my further comments below.

  1. The authors should be consistent in writing “neutralizing epitope” or “neutralization epitope”
  2. The authors have mentioned that they have done statistical analysis. However, I could not see any kind of such analyses or even mentioning the comparison in any legends. Throughout the manuscript like in sentence “In addition, increases in 201 serum ALT levels in group 7 rabbits were significantly greater than increases in group 8 sera at 2 wpc, 202 but not at other wpc times”, when the authors claim significant differences, the P value should be presented.
  3. Fig 2. The graphs should be labelled G1-G6 consistent with the other figures.
  4. Can the authors test which single epitopes(s) is recognized by the antiserum collected from rabbits injected with the peptide comprised of the three peptides?
  5. The authors referred to swine HEV as sHEV once in the entire manuscript (line 256). The authors should be consistent on this.
  6. Fig 4. All rabbits including PBS-injected ones show low levels of viral RNA. Can the authors discuss this in the manuscript? Is this due to contamination of the reagents with viral RNA?
  7. Fig 6. Can the authors validate these results by showing the presence of viral antigen in liver sections with immunoperoxidase staining?

Reviewer 2 Report

The manuscript by Chen et al. studied the protective efficacy of several peptides of HEV genotype 4  capsid against HEV infection in rabbits. They identified  that EPTVKLYTSPSRPF and sp239 provided complete protection, in terms of viremia, liver damage and pathology. 

Overall the paper is clearly written and the results are interesting. 

I only have two minor comments

  1. Please define all error bars and statistics method (for example, 2-sided unpaired Student's t-test, n=? p=? where appropriate) in the figure legends. 
  2. In Table 1, please include the peptide sequence /protein name next to the group numbering. 
  3. Fig.2, please label each subpanel (Group 1, 2...)

Reviewer 3 Report

In the manuscript by Chen et al, the authors examine the potential of using peptides, previously established as neutralizing epitopes, as vaccine candidates against HEV4. The authors did a great job in presenting the experimental design and overall the manuscript is well written. Please see minor comments:

  1. In general, the results are very clear but the display format selected could be improved to ease reading. For example, in Figure 2, the graphs should have tittles indicating the immunizing peptide use per group. 
  2. For Figures 3, it is hard to read the viral RNA data from serum and fecal samples. I will recommend to change the graphing format and include both sets of data in the same graph.
  3. Figure 6, scale bar is shown in all the panels but the measurement is not indicated. 
  4. Also, no where in the manuscript is indicated why it is necessary to have 4 immunizations before the challenge. Clearly use of this vaccine will not be practical if using it for pigs. The authors should discuss this. 

Round 2

Reviewer 1 Report

No further comments.